# A Novel Single-Tube Eicosaplex/Octaplex PCR System for the Detection of Extended-Spectrum β-Lactamases, Plasmid-Mediated AmpC β-Lactamases, and Integrons in Gram-Negative Bacteria

**DOI:** 10.3390/antibiotics12010090

**Published:** 2023-01-04

**Authors:** Ahmed M. Soliman, Hirofumi Nariya, Daiki Tanaka, Toshi Shimamoto, Tadashi Shimamoto

**Affiliations:** 1Department of Microbiology and Immunology, Faculty of Pharmacy, Kafrelsheikh University, Kafr El-Sheikh 33516, Egypt; 2Laboratory of Food Microbiology, Graduate School of Human Life Sciences, Jumonji University, Niiza 352-8510, Japan; 3Laboratory of Food Microbiology and Hygiene, Graduate School of Biosphere Science, Hiroshima University, 1-4-4 Kagamiyama, Higashihiroshima 739-8528, Japan; 4Laboratory of Food Microbiology and Hygiene, Graduate School of Integrated Sciences for Life, Hiroshima University, 1-4-4 Kagamiyama, Higashihiroshima 739-8528, Japan

**Keywords:** multiplex PCR, AmpC, β-lactamase-encoding genes, eicosaplex, octaplex, integrons, Enterobacterales, Gram-negative bacteria

## Abstract

We developed two multiplex polymerase chain reactions (PCRs) for the detection of extended-spectrum β-lactamases (ESBLs), plasmid-mediated AmpC β-lactamases, *aac(6′)-Ib* gene, and integrase genes (*intI1*, *intI2*, and *intI3*) in class 1, 2, and 3 integrons in Gram-negative bacteria. We evaluated the PCRs using 109 Gram-negative isolates from non-organic (ANO) and organic (AO) vegetables and fruits. Screening of ANO substances identified five SHV, one TEM-1, one CTX-M, 20 AmpC-CS, and two *intI1* positives. DNA sequencing revealed CTX-M in *Pantoea* spp. was *bla*_RANH-2_, a plasmid-mediated CTX-M related ESBL gene only found in *Rahnella* spp. Of the 20 AmpC-CS positives, 10 were CMY/MIR/ACT/EC (3 new variants), eight were ACT, one was AZECL, and one was new *Pseudomonas*-related AmpC family. Screening of AO substances identified 11 SHV, two TEM-1, three CTX-M (one OXY-2, two CTX-M-14/-15), two OXA-9, 13 AmpC-CS and one *intI1* positives. The 13 AmpC-CS positives were five CMY/MIR/ACT/EC, three ACT, one MOX-12 variant, and four ADC (one ADC-25 and three new variants). We developed a rapid, easy-to-perform, low-cost, and reliable multiplex PCR system for screening clinically relevant β-lactamases and integrons in Gram-negative bacteria. We showed the prevalence of ESBLs and AmpC β-lactamases among our panel of ampicillin-resistant Gram-negative strains and detection of NDM and OXA carbapenemases.

## 1. Introduction

Antimicrobial resistance (AMR) is defined as the ability of microorganisms (bacteria or fungi) to overcome the action of the antibiotics that intend to kill them (i.e., bactericidal), or inhibit their growth (i.e., bacteriostatic) [1]. AMR is considered one of the greatest threats to global public health [1]. In the USA, annual data estimate more than 2.8 million infections with antibiotic-resistant bacteria, causing more than 35,000 deaths [1]. Because of antimicrobial-resistant infections, it is estimated that approximately 10 million global deaths, with fiscal losses of > USD 100 trillion, will occur by 2050 [2].

β-lactams are one of the most clinically important and widely prescribed classes of antibiotics used for the treatment of Gram-negative or Gram-positive bacteria [3]. The major mechanism of β-lactam resistance is the production of β-lactamases [4]. Extended-spectrum β-lactamases (ESBLs) and plasmid-mediated AmpC β-lactamases are groups of these enzymes that should be considered by clinical microbiologists [5,6]. AmpC β-lactamases have been identified globally in nosocomial and non-nosocomial bacterial strains with activity against penicillin, cephalosporin, cephamycins (e.g., cefoxitin and cefotetan), oxyiminocephalosporins (e.g., ceftriaxone, cefotaxime, and ceftazidime), and monobactam (e.g., aztreonam) [5]. ESBLs have been associated with outbreaks and have become a key cause of hospital-acquired infections, especially in intensive care units [6,7]. Recently, the most frequently encountered ESBLs in clinical, environmental, and animal sources are TEM, SHV, and CTX-M [6,7].

Integrons are DNA elements found in many bacterial species and are characterized by their ability to capture, exchange, and express small mobile elements called gene cassettes [8]. Cassettes usually contain only one open reading frame, possibly any gene, and an *attC* recombination site [8]. A specialized IntI site-specific recombinase encoded by the integron recognizes *attC* and incorporates cassettes into an *attI* site located adjacent to the *intI* gene [8]. An integron can comprise zero to hundreds of cassettes due to repeated *attC*-*attI* recombination [8]. Therefore, integrons participate in increasing the plasticity of bacterial, plasmid, and genomes and facilitating the widespread distribution of genetic materials among bacteria [8]. Three integron types are classified according to the encoded integrase enzyme into class 1, class 2, and class 3 integrons and are usually associated with AMR genes [8,9].

Whole-genome sequencing (WGS) is the best way to identify all known AMR genes in bacteria; however, it is expensive for epidemiological surveys [10]. For example, ResFinder, a web-based tool available at the Center for Genomic Epidemiology (https://cge.cbs.dtu.dk/services/ResFinder/) (accessed on 26 November 2022), is widely used to identify AMR genes in WGS data [10]. Several interesting multiplex PCR systems using specific primer sets have been published and used for screening ESBLs or plasmid-mediated AmpC β-lactamases among *Enterobacteriaceae* and other Gram-negative bacteria [11,12,13,14]. Unfortunately, these primers could not fully cover the increasing number of alleles deposited in the GenBank database. In this study, we aimed to develop a multiplex PCR assay for the detection of all variants of i) ESBLs (CTX-M, TEM, SHV, OXA-1-like, OXA-2-like, OXA-5-like, OXA-9-like) and ii) AmpC β-lactamases (ACC, ACT/MIR, DHA, CMY/LAT, FOX, MOX, ADC, PDC) submitted to GenBank as of May 2018 in combination with class 1, 2, and 3 integrons.

## 2. Materials and Methods

### 2.1. Bacterial Isolates

For optimization of the two multiplex PCRs, previously well-characterized strains that had been recovered from different parts of the world were used as positive controls (Table 1) [15,16,17,18,19,20]. Owing to the lack of positive control strains for *bla*_OXA-2_, *bla*_FOX_, *bla*_ACC_, and integrase genes of class 3 integrons (*intI3*), the corresponding gene was synthesized by Eurofins Genomics KK (Tokyo, Japan) and incorporated into the kanamycin resistant (Km^R^) plasmid pEX-K4J2 (https://eurofinsgenomics.jp/media/29197/pex-k4j2vector-mapseq.pdf) (accessed on 10 November 2022), which was further electroporated into *E. coli* DH10B (Table 2).

For evaluation, 109 Gram-negative strains were grown on MacConkey agar plates containing 100 μg/mL ampicillin (AMP) from October 2014 to August 2015 from 27 non-organic (ANO) (*n* = 54 isolates) and 21 organic (AO) (*n* = 55 isolates) vegetables and fruits (including imported ones) retailed in Hiroshima Prefecture, Japan. The 16S rRNA gene was amplified and sequenced using 27F primer (5′-AGAGTTTGATCMTGGCTCAG-3′) [21] and 1492R (5′-CGGYTACCTTGTTACGACTT-3′). The strains were further screened on Luria-Bertani (LB) (Lennox) agar medium supplemented with 1 and 4 μg/mL meropenem (MEM), 4 μg/mL cefotaxime (CTX), and 16 μg/mL ceftazidime (CAZ).

### 2.2. Design of Specific Primers for Eicosaplex and Octaplex PCRs

Two multiplex PCRs were developed in this study: (1) eicosaplex PCR for the detection of ESBLs (*bla*_CTX-M_, *bla*_OXA-1_, *bla*_OXA-2_, *bla*_OXA-5_, *bla*_OXA-9_, *bla*_SHV_, *bla*_TEM_), plasmid-mediated AmpC β-lactamases (*bla*_ACC_, *bla*_ACT/MIR_, *bla*_CMY/LAT_, *bla*_DHA_, *bla*_FOX_, and *bla*_MOX_ family plus intrinsic *bla*_ADC_ and *bla*_PDC_ family in *Acinetobacter* and *Pseudomonas*, respectively) families, in combination with integrase genes (*intI1*, *intI2*, and *intI3*) in class 1, 2, and 3 integrons and *aac(6’)-Ib* gene in Gram-negative bacteria and (2) octaplex PCR for differentiation of plasmid-mediated AmpC β-lactamases (*bla*_ACC_, *bla*_ACT/MIR_, *bla*_CMY/LAT_, *bla*_DHA_, *bla*_FOX_, *bla*_MOX_, *bla*_ADC_ and *bla*_PDC_). The *bla*_AmpC_ in the eicosaplex PCR assay targets the AmpC conserved (CS) region (amplicon size, approximately 374 bp) using each family-specific AmpC-CS primer set. AmpC-CS positives were then distinguished by octaplex PCR using a family-specific (SP) primer set, AmpC-SP. In the eicosaplex PCR, amplification of *bla*_ADC_ and *bla*_ACC_ resulted in a 388 bp product, while amplification of *bla*_MOX_ and *bla*_FOX_ resulted in a 371-bp product.

DNA sequences of the different genes and their variants were obtained from the GenBank database and aligned using Clustal Omega software (Clustal W) (https://www.ebi.ac.uk/Tools/msa/clustalo/) (accessed on 10 November 2022). Specific primers were manually designed for each family to cover all alleles deposited in the GenBank database as of May 2018 and amplify internal fragments of different sizes (Table 3 and Table 4, Figure 1 and Figure 2). For example, CTX-M primers were designed to cover all 214 publicly available CTX-M variants (including those belonging to phylogenetic group 1, 2, 8, and 9), KLUA, KLUC, KLUG, and TOHO-1 to TOHO-3.

### 2.3. Eicosaplex and Octaplex PCR Technique

DNA was prepared using the boiling lysate method, as previously reported [15]. Total DNA (0.2 μL) was subjected to eicosaplex or octaplex PCR in a 10 μL reaction mixture containing 5 μL of 2 X Gflex PCR Buffer (TaKaRa Bio, Shiga, Japan, Cat. #R060A/B) (2 mM Mg^2+^, ~400 μM dNTP plus), varying concentrations of the specific primers (Table 3 and Table 4), 4.2 μL Milli-Q water (4.6 μL in case of octaplex PCR) and 0.1 μL Tks Gflex™ DNA Polymerase (1.25 units/μL) (TaKaRa Bio, Shiga, Japan, Cat. #R060A/B). Thermal cycling was performed as follows: initial denaturation at 98 °C for 1 min; 35 cycles of 98 °C for 10 s, 50 °C for 15 s, and 68 °C for 30 s; and a final elongation at 68 °C for 15 s. For octaplex PCR, primer annealing was optimized at 57.5 °C. The PCR products were imaged after running at 100 V for 60 min on a 3% agarose gel and stained with a solution containing ethidium bromide.

### 2.4. DNA Sequencing and Analysis

Amplified genes were purified using the ExoSAP-IT™ PCR product cleanup kit (https://assets.fishersci.com/TFS-Assets/LSG/manuals/78200b.pdf) (accessed on 10 November 2022) (Applied Biosystems, Foster City, CA, USA) and sequenced in both directions using a 3730xl DNA Analyzer (Applied Biosystems). To identify genes or their variants, a search was performed using the BLAST program accessible on the NCBI BLAST homepage (https://blast.ncbi.nlm.nih.gov/Blast.cgi) (accessed on 10 November 2022).

### 2.5. Whole Genome Sequencing and Analysis

Some interesting carbapenem-resistant Gram-negative bacteria detected in this study were subjected to WGS using Illumina MiniSeq (Illumina, San Diego, CA, USA) and Oxford MinION Nanopore (Oxford Nanopore Technologies, Oxford, UK). Hybrid assembly was performed as previously reported to produce high-quality sequences [22]. The complete genome sequences of the three isolates and their analyses have been previously published [20].

## 3. Results and Discussion

### 3.1. MER, CTX, CAZ-Resistant Strains from the ANO and AO Substances

We screened 109 strains isolated in this study on LB agar medium supplemented with 4 μg/mL MEM, 4 μg/mL CTX, and 16 μg/mL CAZ. A total of 8, 33, and 15 different Gram-negative strains were resistant to MEM 4, CTX 4, and CAZ 16, respectively, indicating the potential for spreading and infecting human beings. Fresh produce is commonly consumed raw or is under cooked [23]. Subsequently, a significant portion of the latest foodborne outbreaks have been associated with fresh produce polluted by bacteria [24]. Similarly, the spread of antimicrobial resistance in humans can occur through the consumption of vegetables contaminated with multidrug-resistant bacteria [24].

### 3.2. Design of Specific Primers for Eicosaplex and Octaplex PCRs

In this study, we developed two multiplex PCRs, namely eicosaplex and octaplex, for the detection of ESBLs and AmpC β-lactamases in combination with integrase genes (*intI1*, *intI2*, and *intI3*) in class 1, 2, and 3 integrons and *aac(6’)-Ib* gene in Gram-negative bacteria. After the optimization of the PCR reaction conditions, the expected target size of each amplicon was obtained using either the primer pairs in a simplex (Figure 3C,D) or multiplex (Figure 3A,B) approach. The specificity was good, with amplification of all the expected fragments (Figure 3). For example, every DNA template of *bla*_TEM_ and *bla*_CTX-M_ resulted in 602 bp and 509 bp, respectively.

### 3.3. Evaluation of the Eicosaplex and Octaplex PCRs with Gram-Negative Strains Isolated from Vegetables and Fruits in Japan

To verify the specificity of the system, simplex and multiplex PCRs were performed on 109 Gram-negative strains isolated in this study. All the PCR products were bidirectionally sequenced.

ESBLs were identified in 25 isolates (23%), plasmid-mediated AmpC β-lactamases were detected in 33 isolates (30%), and 3 isolates (2.75%) carried both a plasmid-mediated AmpC β-lactamase and a class 1 integron, from which another isolate additionally carried the broad-spectrum β-lactamase TEM-1.

Screening of ANO substances identified five SHV (one SHV-1, three SHV-1 variants, and one SHV-41 variant), one TEM-1, one CTX-M, 20 AmpC-CS, and two *intI1* positives (Table 5 and Table 6). Furthermore, DNA sequence analysis revealed that CTX-M in *Pantoea* spp. was *bla*_RANH-2_, a plasmid-mediated CTX-M-related ESBL gene found only in *Rahnella* spp. The 20 AmpC-CS positives were 10 CMY2/MIR/ACT/EC (three new variants), eight ACT (two ACT-16, two ACT-61, one ACT-64, two ACT-32, and one ACT-51 variants), one AZECL, and one new *Pseudomonas*-related AmpC family. Screening of the AO identified 11 SHV (two SHV-1, one SHV-27, one SHV-11, one SHV-11 variant, one LEN-13, two LEN-19 variants, one LEN-27 variant, one LEN-25, and one LEN-16), two TEM-1, three CTX-M (one OXY-2, two CTX-M-14/-15), two OXA-9, two *aac(6′)-Ib*, 13 AmpC-CS, and one *intI1* positive. The 13 AmpC-CS positives were five CMY2/MIR/ACT/EC (one new variant), three ACT (one ACT-51 variant, one ACT-2 variant, and one ACT-9), one MOX-12 variant, and four ADC (one ADC-25 and three new variants).

CMY2/MIR/ACT/EC and ACT enzymes were the most frequently detected plasmid-mediated AmpC β-lactamases (found in 15 and 11 isolates, respectively); only one variant of MOX-12 AmpC β-lactamase was reported. Carbapenemase-encoding genes were detected using previously published primers and conditions [25]. New Delhi metallo-β-lactamase 1 (NDM-1) was detected in two *K. pneumoniae* (AO15 and AO22) isolates in this study [20]. The two isolates belonged to the same clone and sequence type 15 (ST15) [20]. Complete genome sequencing and ResFinder (https://cge.food.dtu.dk/services/ResFinder/) (accessed on 10 November 2022) analysis identified that both the isolates carried 19 different antimicrobial resistance genes. *bla*_NDM-1_ was carried by a self-conjugable IncFII(K):IncR plasmid of 122,804 bp with other genes conferring resistance to β-lactams (*bla*_CTX-M-15_, *bla*_OXA-9_, and *bla*_TEM-1A_), fluoroquinolones [*aac(6′)-Ib-cr*], aminoglycosides [*aac(6′)-Ib*, *aadA1*, *aph(3′)-VI*], and quinolones (*qnrS1*) [20]. In addition, another *A. baumannii* AO22 isolate identified in our study carried a chromosomal *bla*_OXA-66_ and two copies of *bla*_OXA-72_ on a 10,880 bp GR2-type plasmid [20]. None of the carbapenemase genes targeted were found in this study in the other five isolates showing decreased susceptibility to meropenem. A complete description of the characteristics of the 109 Gram-negative isolates in this study are presented in Table 7 and Table 8.

A recent study from Nepal reported *E. coli* and *Salmonella* producing multidrug-resistant ESBL in vegetable salads served at restaurants and hotels. Alarmingly, three samples harbored *E. coli* O157: H7 [26]. Colosi et al. (2020) reported β-lactamase-producing Enterobacterales in 7.9% of raw vegetables retailed in Romania, with 5.5% showing the ESBL or AmpC phenotype and 2.4% associated with carbapenemase producers [27]. Therefore, retail vegetables and fruits might be important reservoirs of multidrug-resistant bacteria that produce ESBLs, AmpC, or carbapenemases. Furthermore, the detection of these organisms in fresh vegetables in Japan, a country with quite low levels of antimicrobial resistance and high-level sanitary standards, is disturbing and poses food safety and public health concerns, as resistant organisms might be transmitted to humans. From a global health perspective, surveillance plans using a fast, low-cost, and effective method (i.e., eicosaplex/octaplex system) are essential at both the international and local levels to offer evidence for risk improvement strategies to reduce the spread of resistance.

The PCR products can be sequenced directly and efficiently, permitting the detection of a large number of β-lactamases. Even if the strain carried two different variants of the same β-lactamase, it could be detected by sequence analysis of the resulting amplicon (for example, the positive control strains LM22-1 carrying *bla*_CTX-M-9_ and *bla*_CTX-M-15_, and AO15/22 carrying *bla*_CTX-M-14b_ and *bla*_CTX-M-15_). Here, we developed two rapid, easy-to-perform, cost-effective, and reliable multiplex PCRs for the efficient screening of ESBLs, AmpC β-lactamases, and integrons in Gram-negative bacteria. This method allows for the detection of new variants of ESBLs and AmpC β-lactamases. Interestingly, in eicosaplex PCR, we were able to amplify 20 targets in a single tube using Tks Gflex DNA Polymerase (TaKaRa Bio, Shiga, Japan, Cat. #R060A/B).

## 4. Conclusions

In this study, we developed two multiplex PCRs, eicosaplex and octaplex, in a single tube (using Tks Gflex™ DNA Polymerase), for the detection of ESBLs, plasmid-mediated AmpC β-lactamases, and class 1, 2, and 3 integrons in Gram-negative bacteria. This PCR assay is a rapid, low-cost, easy-to-perform, and reliable method for screening clinically relevant mobile and disseminated β-lactamases and integrons. This method allows for the detection of new variants of ESBLs and AmpC β-lactamases. For epidemiological surveys in low- and middle-income countries, and for reference laboratories, our technique may be extremely helpful in reducing the time and cost of multiplex PCRs. This PCR system will assist in controlling the emergence and spread of frequently encountered β-lactamases. In conclusion, our study showed the prevalence of ESBLs and AmpC among our panel of ampicillin-resistant Gram-negative strains isolated from vegetables and fruits in Japan, and the detection of NDM and OXA carbapenemases. This situation is disturbing and poses food safety and public health concerns, as resistant organisms can be transmitted to humans. This multiplex PCR assay is a promising workflow in bacterial diagnosis in high-risk patients providing a great assistance in optimizing and fast choice of antibiotics for treating infections due to ESBLs- and AmpC β-lactamases-producing Gram-negative pathogens.

## Figures and Tables

**Figure 1 antibiotics-12-00090-f001:**
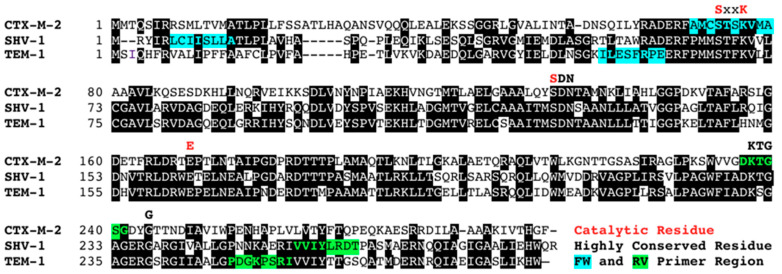
Example of the primer design for eicosaplex PCR. The catalytic residues for β-lactamases are indicated above the alignment of the amino acid sequences of each β-lactamase family with red color. The forward and reverse primers are highlighted in turquoise and bright green colors.

**Figure 2 antibiotics-12-00090-f002:**
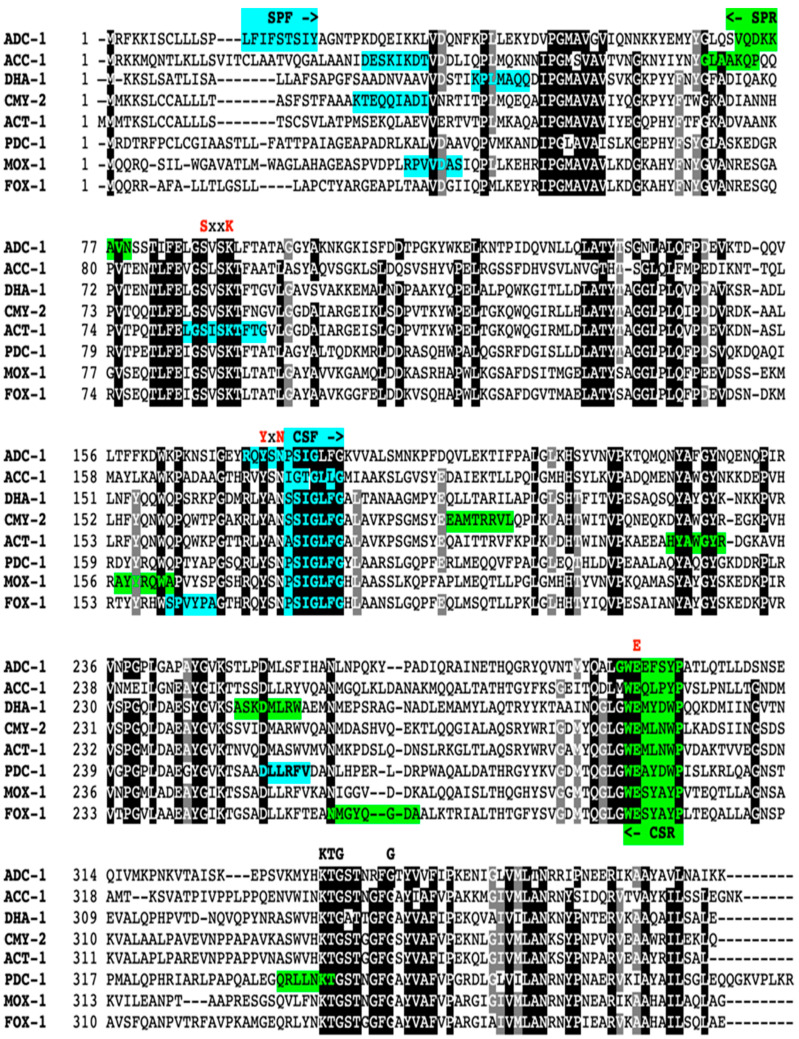
Example of the primer design for octaplex PCR for the major AmpC families. The catalytic residues for β-lactamases are indicated above the alignment of the amino acid sequences of each β-lactamase family with a red color. The forward and reverse primers are highlighted in turquoise and bright green colors, respectively. SPF and SPR indicates the specific forward and reverse primers for each AmpC family in the octaplex, respectively. CSF and CSR indicate the conserved forward and reverse primers for each AmpC family in the eicosaplex PCR, respectively.

**Figure 3 antibiotics-12-00090-f003:**
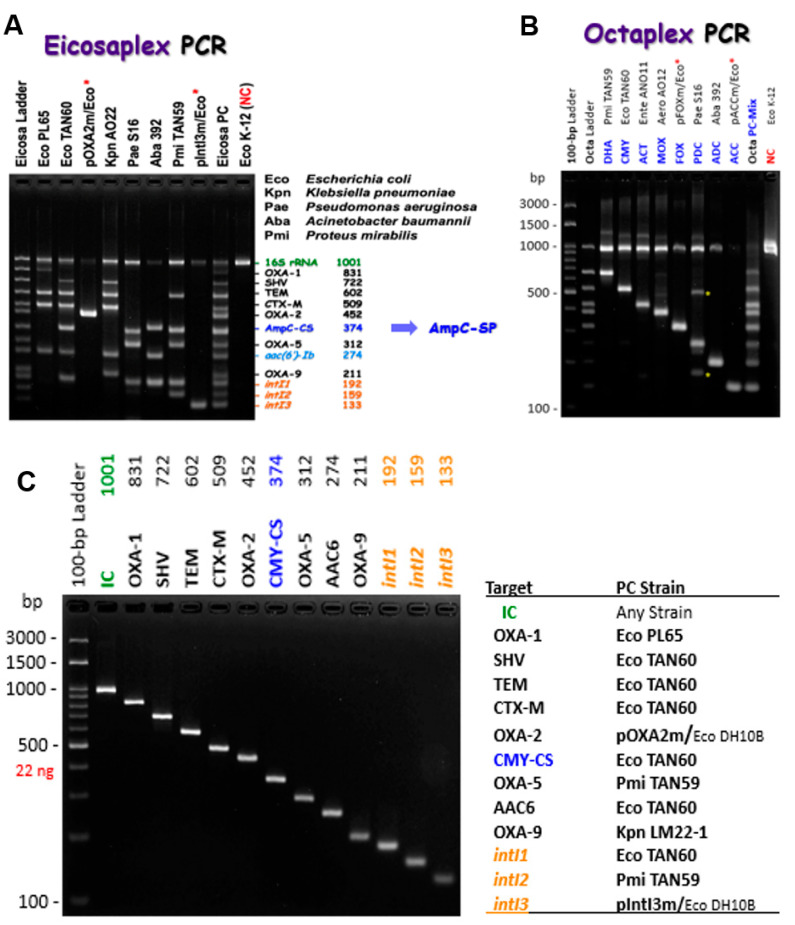
Amplification of the expected amplicons of ESBLs, AmpC β-lactamases, and the integrase genes of class 1, 2 and 3 integrons, from the positive control strains and plasmids used in this study. (A) indicates the eicosaplex PCR which amplifies *bla*_CTX-M_, *bla*_OXA-1_, *bla*_OXA-2_, *bla*_OXA-5_, *bla*_OXA-9_, *bla*_SHV_, *bla*_TEM_, AmpC-CS (*bla*_ACC_, *bla*_ACT/MIR_, *bla*_CMY/LAT_, *bla*_DHA_, *bla*_FOX_ and *bla*_MOX_, *bla*_ADC_, and *bla*_PDC_), *aac(6’)-Ib*, and integrase genes (*intI1*, *2* and *3).* In case of obtaining a band of AmpC-CS (374 bp) in (**A**), we should use (**B**) which indicates the octaplex PCR amplifying and differentiating specifically between *bla*_ACC_, *bla*_ACT/MIR_, *bla*_CMY/LAT_, *bla*_DHA_, *bla*_FOX_, *bla*_MOX_, *bla*_ADC_, and *bla*_PDC_. The PCR products were separated in a 3% agarose gel. The lanes were named according to the positive control strain or plasmid used, for example, lane Eco PL65 indicates the amplification of *bla*_OXA-1_, *bla*_TEM-1_, *bla*_CTX-M-15_, and *aac(6′)-Ib-cr.* Lane pOXA2m/Eco indicates the amplification of the synthesized *bla*_OXA-2_. Lane Eicosa and Octa Ladders indicate a single amplification of each target in the two multiplexes, as shown in (**C**,**D**), then all the amplicons of each one were mixed in the same ratio and loaded into the agarose gel. NC indicates the negative control strain *E. coli* K12. Red asterisks indicate synthesized genes inserted in the pEX-K4J2 (Km^R^) plasmid. Yellow asterisks indicate non-specific band that appeared only from *P. aeruginosa* in the case of octaplex PCR. Blue colored text indicates AmpC β-lactamases. Orange colored text indicates integrons (*intI1*, *intI2*, and *intI3*). Green colored text indicates Internal controls (IC).

**Table 1 antibiotics-12-00090-t001:** Positive control bacteria used in this study.

Strain Name	*Bacterial* spp.	Source	Country	Isolation Year	Resistance Genes	Reference
Eco TAN60	*E. coli*	Clinical	Egypt	2014	*bla*_NDM-1_, *bla*_SHV-12_, *bla*_CTX-M-15_, *bla*_TEM-1_, *bla*_CMY-2_, class 1 integron (*aac(6’)-Ib*), *qnrS*, *rmtC*	[19]
Eco TAN92	*E. coli*	Clinical	Egypt	2014	*bla*_NDM-1_, *bla*_CTX-M-15_, *bla*_CMY-2_, *rmtC*, class 1 integron carrying (*aac(6´)-Ib*)	[19]
Eco PL65	*E. coli*	Clinical	Palestine	2006	*bla*_TEM-1_, *bla*_OXA-1_, *bla*_CTX-M-15_, *aac(6´)-Ib-cr*	[16]
Pmi TAN59	*P. mirabilis*	Clinical	Egypt	2014	class 1 integron (*aacA5*-*aadA7*-*qacE1*- *sul1*), *bla*_TEM-1_, *bla*_DHA_, *bla*_OXA-5_, *qnrA1*, class 2 integron (*intI2*)	[15]
Kpn D6	*K. pneumoniae*	Water	Japan	2016	*bla*_TEM-1_, *bla*_SHV-12_, *bla*_OXA-1_, *bla*_CTX-M-15_, *aac(6´)-Ib-cr5*, *oqxAB*, class 1 integron	This study
Kpn LM22-1	*K. pneumoniae*	Chicken	Japan	2016	*bla*_SHV-71_, *oqxAB*, *fosA*, *bla*_VIM-1_, *mcr-9*, *bla*_TEM-1B_, *bla*_CTX-M-9_, *aadA24*, *ant(2´´)-Ia*, *aadA2*, *aph(3´)-Ia*, *sul1*, *dfrA1*, *bla*_NDM-1_, *aac(6´)-Ib*, *aadA1*, *bla*_CTX-M-15_, *bla*_OXA-9_, *bla*_TEM-1A_, *qnrS1*, *aac(6´)-Ib-cr*	[18]
Kpn AO15/22	*K. pneumoniae*	Vegetable	Japan	2015	*bla*_SHV-28_, *fosA*, *oqxAB*, *bla*_SHV-1_, *bla*_TEM-1A_, *tet*(D), *aac(6’)-Ib*, *aadA1*, *aph(3’)-VI*, *bla*_CTXM-15_, *bla*_NDM-1_, *bla*_OXA-9_, *aac(6’)-Ib-cr*, *qnrS1*, *aph(6)-Id*, *aph(3’)-VIb*, *aph(3’’)-Ib*, *bla*_CTXM-14b_	[20]
Pst TAN3	*P. stuartii*	Clinical	Egypt	2014	class 1 integron (*aadA2*-*lnuF*), *bla*_TEM-1_, *bla*_DHA-1_, *qnrD*, *floR*, class 2 integron (*intI2*)	[17]
Pst TAN14	*P. stuartii*	Clinical	Egypt	2014	*bla*_NDM-1_, *bla*_CMY-2_, *bla*_DHA-1_, *rmtC*, *qnrA*, *qnrD*, class 1 integron carrying (*aac(6’)-Ib*)	[19]
Pae S16	*P. aeruginosa*	Clinical	Egypt	2014	class 1 integron (*bla*_VIM-24_-*aadB*-*bla*_OXA-10_), *bla*_PDC_	[19]
Aba 392	*A. baumannii*	Clinical	Egypt	2015	*bla*_OXA-23-like,_*bla*_OXA-51-like_, class 1 integron, *bla*_ADC,_*aac(6’)-Ib*	This study
Aba AO22	*A. baumannii*	Vegetable	Japan	2015	class 1 integron, *bla*_ADC-25_, *bla*_OXA-66_, *sul2*, *tet*(B), *aac(3)-Ia*, *aac(6’)-Ip*, *aph(3’’)-Ib*, *aph(6)-Id*	[20]
Aero AO12	*Aeromonas* spp.	Vegetable	Japan	2014-2015	*bla* _MOX-12_	This study
Aero 19-2B	*Aeromonas* spp.	Vegetable	Japan	2019	*bla* _FOX_	This study
Ente ANO11	*Enterobacter* spp.	Vegetable	Japan	2014-2015	*bla* _ACT-64_	This study

**Table 2 antibiotics-12-00090-t002:** Positive control plasmids used in this study.

Plasmid Name	Vector	Resistance Genes	Reference
pOXA-2m	pEX-K4J2 (Km^R^)	*bla* _OXA-2_	This study
pIntI3m	pEX-K4J2 (Km^R^)	Integrase gene of class 3 integron (*intI3*)	This study
pFOXm	pEX-K4J2 (Km^R^)	*bla* _FOX_	This study
pACCm	pEX-K4J2 (Km^R^)	*bla* _ACC_	This study

All the pEX-K4J2 (Km^R^) plasmids were transformed into *E. coli* DH10B.

**Table 3 antibiotics-12-00090-t003:** Group-specific oligonucleotides used in this study for eicosaplex PCR.

Primer	Sequence (5’–3’) ^1^	Targeted Gene or Targeted β-Lactamase(s) ^2^	Amplicon Size (bp)	Primer Concentration (pmol/μL)	Reference
515F 1492R	GTGCCAGCMGCCGCGGTAA CGGYTACCTTGTTACGACTT	16S rRNA	1001	0.2 0.2	This study
OXA1-WGF OXA1-WGR	ATGAAAAACACAATACATATCAACTTC TTATAAATTTAGTGTGTTTAGAATGGTG	OXA-1, OXA-4, OXA-30, **OXA-31**, OXA-47, **OXA-224**, OXA-320, OXA-392, OXA-543	831	0.4 0.4	This study
SHV-FW SHV-RV	GTGTATTATCTCCCTGTTAGCC GGCCAAGCAGGGCGACAAT	SHV-1 to SHV-203	722	0.2 0.2	This study
TEM-FW TEM-RV	TTGAGAGTTTTCGCCCCGAA ACGGGAGGGCTTACCATCTG	TEM-1 to TEM-232	602	0.1 0.1	This study
CTXM-FW CTXM-RV	TGCAGYACCAGTAARGTKATGGC CCGCTGCCGGTYTTATC	CTX-M-1 to CTX-M-214, KLUA, KLUC, KLUG, and TOHO-1 to TOHO-3	509	0.3 0.3	This study
OXA2-FW OXA2-RV	ATAGTTGTGGCAGACGAACG TTGACCAAGCGCTGATGTTC	OXA-2, OXA-3, **OXA-15**, OXA-20, OXA-21, **OXA-32**, OXA-37, OXA-46, **OXA-53**, OXA-118, OXA-119, **OXA-141**, **OXA-161**, **OXA-210**, **OXA-226**, **OXA-415**, **OXA-539**, **OXA-540**, **OXA-541**, OXA-543, OXA-544	452	0.2 0.2	This study
CMY/LAT-CSFCMY/LAT-CSR	TCCAGCATTGGTCTGTTTGG GGCCAGTTCAGCATCTCCCA	CMY-2 to CMY-157, BIL-1, LAT-1 to LAT-4, CFE-1	374	0.1 0.1	This study
MIR/ACT-CSFMIR/ACT-CSR1 MIR/ACT-CSR2	GCCAGCATCGGTCTTTTTGG GGCCAGTTGAGCATCTCCCA GGCCAGTTTAGCATTTCCCA	ACT-1 to ACT-54, MIR-1 to MIR-22	374	0.1 0.1 0.1	This study
DHA-CSF DHA-CSR	AGCAGTATCGGCCTGTTTGG GGCCAGTCATACATTTCCCA	DHA-1 to DHA-25	374	0.1 0.1	This study
MOX-CSF1 MOX-CSF2 MOX-CSR	CCCAGCATAGGGCTGTTCGG CCCAGCATCGGGCTCTTTGG GGATAGGCGTAACKCTCCCA	MOX-1 to MOX-13, CMY-1, CMY-8, CMY-8b, CMY-9, CMY-10, CMY-11, CMY-19	371	0.1 0.1 0.1	This study
FOX-CSF FOX-CSR	CCCAGCATMGGCCTGTTTGG GGATAGGCGTARCTCTCCCA	FOX-1 to FOX-16	371	0.1 0.1	This study
PDC-CSF PDC-CSR	CCGAGCATCGGYCTGTTCGG GGCCAGTCGTAGGCTTCCCA	PDC-1 to PDC-249	374	0.1 0.1	This study
ACC-CSF ACC-CSR	ATCGGTACYGGTTTGCTAGG GGATATGGCARCTGCTCCCA	ACC-1 to ACC-7	380	0.1 0.1	This study
ADC-CSF ADC-CSR	GACAATATTCAAAYCCAAGYATTGG GGATAAGAAAAYTCTTCCCAACC	ADC-1 to ADC-107	388	0.4 0.4	This study
OXA5-FW OXA5-RV	GTATTTCAACAAATYGCCAGAGA CCACCAWGCGACACCAGGA	OXA-5, OXA-7, OXA-10, **OXA-11**, OXA-13, **OXA-14**, **OXA-17**, **OXA-19**, **OXA-28**, **OXA-35**, OXA-56, **OXA-74**, OXA-101, OXA-129, OXA-142, **OXA-145**, **OXA-147**, **OXA-183**, OXA-233, **OXA-240**, OXA-246, OXA-251, OXA-256, OXA-368, OXA-454, OXA-520	312	0.4 0.4	This study
AAC6-FW AAC6-RV	TTGCGATGCTCTATGAGTGGCTA AGTTGTGATGCATTCGCCAG	*aac(6′)*-*Ib*	274	0.1 0.1	This study
OXA9-FW OXA9-RV	CAGTTCCGTGGCTTCTGATG GTTGTATTCCGGCTTCAATTCC	OXA-9a, OXA-9b	211	0.1 0.1	This study
IntI1-FW IntI1-RV	AGCTTGGCACCCAGCCTG GACACCGCTCCGTGGATC	*intI1* of class 1 integron	192	0.15 0.15	This study
IntI2-FW IntI2-RV	AAGGTTATGCGCTGAAAACTGAA TCTGCGTGTTTATGGCTACATG	*intI2* of class 2 integron	159	0.1 0.1	This study
IntI3-FW IntI3-RV	CACCGAGAAGCAAGTGG AATCCGCTTGCGTTCTG	*intI3* of class 3 integron	133	0.2 0.2	This study

^1^ R = G or A, Y = C or T, M = A or C, K = G or T, W = A or T. ^2^ Bolded OXA β-lactamases indicate extended-spectrum β-lactamase activity.

**Table 4 antibiotics-12-00090-t004:** Group-specific oligonucleotides used in this study for octaplex PCR.

Primer	Sequence (5’–3’) ^1^	Targeted Gene or Targeted β-Lactamase(s)	Amplicon Size (bp)	Primer Concentration (pmol/μL)	Reference
DHA-SPF DHA-SPR	ACCGCTGATGGCACAGCAG CAGCGCAGCATATCTTTTGAG	DHA-1 to DHA-25	648	0.2 0.2	This study
CMY/LAT-SPFCMY/LAT-SPR1 CMY/LAT-SPR2	AAAACAGAACAACARATTGCCGATA GGACGCGTCTGGTCATTGCC GGACGCGGGTGGTCATCGCC	CMY-2 to CMY-157, BIL-1, LAT-1 to LAT-4, CFE-1	529	0.2 0.2 0.2	This study
MIR/ACT-SPF1MIR/ACT-SPF2MIR/ACT-SPR1MIR/ACT-SPR2	CTGGGYTCTATAAGTAAAACCTTCACCG CTGGGCTCAATCAGCAAAACCTTCACCG CGGTATCCCCAGGCGTAATG CGATAGCCCCAGGCGTAATG	ACT-1 to ACT-54, MIR-1 to MIR-22	428	0.3 0.3 0.3 0.3	This study
MOX-SPF MOX-SPR	GCCCCGTGGTGGATGCCAG GYCCACTGGCGGTAGTAGGC	MOX-1 to MOX-13, CMY-1, CMY-8, CMY-8b, CMY-9, CMY-10, CMY-11, CMY-19	391	0.2 0.2	This study
FOX-SPF FOX-SPR	TGGTCACCGGTTTATCCGGC GCATCTCCCTGATACCCCATGTT	FOX-1 to FOX-16	323	0.2 0.2	This study
PDC-SPF PDC-SPR	GACCTGCTGCGCTTCGTC GGTCTTGTTCAGCAGGCGCT	PDC-1 to PDC-249	263	0.2 0.2	This study
ADC-SPF ADC-SPR	CTTTTTATTTTTAGTACCTCAATTTATGC TGCTATTTACGGCTTTTTTATCTTGAAC	ADC-1 to ADC-107	202	0.4 0.4	This study
ACC-SPF ACC-SPR	GATGAGAGCAAAATTAAAGACACCG AGGCTGTTTTGCCGCTAACC	ACC-1 to ACC-7	143	0.2 0.2	This study

^1^ R = G or A, and Y = C or T.

**Table 5 antibiotics-12-00090-t005:** Sum of positive strains of the eicosaplex/octaplex PCR system from the ANO substances.

54 ANO	Number of Isolates	CTX-M	OXA-1	OXA-2	OXA-5	OXA-9	SHV	TEM	*aac(6’)*	*intI1*	*intI2*	*intI3*	AmpC	DHA	CMY/LAT	ACT/MIR	MOX	FOX	PDC	ADC	ACC	ND
*Enterobacter*	16	-	-	-	-	-	1	2	-	-	-	-	16	-	-	15	-	-	-	-	-	1
*Klebsiella*	6	-	-	-	-	-	5	-	-	-	-	-	1	-	-	1	-	-	-	-	-	-
*Citrobacter*	2	-	-	-	-	-	-	-	-	-	-	-	2	-	-	-	-	-	-	-	-	2
*Pseudomonas*	1	-	-	-	-	-	-	-	-	-	-	-	1	-	-	-	-	-	-	-	-	1
*Rhanella*	1	1	-	-	-	-	-	-	-	-	-	-	-	-	-	-	-	-	-	-	-	-
Total	26	1	-	-	-	-	6	2	-	-	-	-	20	-	-	16	-	-	-	-	-	4

**Table 6 antibiotics-12-00090-t006:** Sum of positive strains of the eicosaplex/octaplex PCR system from the AO substances.

55 AO	Number of Isolates	CTX-M	OXA-1	OXA-2	OXA-5	OXA-9	SHV	TEM	*aac(6’)*	*intI1*	*intI2*	*intI3*	AmpC	DHA	CMY/LAT	ACT/MIR	MOX	FOX	PDC	ADC	ACC	ND
*Klebsiella*	18	8	-	-	-	2	11	2	2	-	-	-	1	-	-	-	-	-	-	-	-	1
*Enterobacter*	7	-	-	-	-	-	-	-	-	-	-	-	7	-	-	7	-	-	-	-	-	-
*Acinetobacter*	4	-	-	-	-	-	-	-	-	1	-	-	4	-	-	-	-	-	-	4	-	-
*Aeromonas*	1	-	-	-	-	-	-	-	-	-	-	-	1	-	-	-	1	-	-	-	-	-
*Proteus*	1	1	-	-	-	-	-	-	-	-	-	-	-	-	-	-	-	-	-	-	-	-
Total	31	9	-	-	-	2	11	2	2	1	-	-	13	-	-	7	1	-	-	4	-	1

**Table 7 antibiotics-12-00090-t007:** Features of the Gram-negative isolates grown on MacConkey plates containing 100 μg/mL ampicillin from 21 organic (AO: 55 isolates) vegetables and fruits.

No.	Strain AO	Source (# Imported)	Identification by 16S rRNA Sequencing ^2^	Growing on LB Agar Medium Supplemented with ^3^					Genotype Eicosa (Octa) Plex PCR ^4^
				AMP 100 24 h	MER 4 24 h	MER 1 24 h	CTX 4 24 h	CAZ 16 24 h	
1	1-1	Green onion	ND	+					ND
2	1-2		ND	+					ND
3	2-1	Hakusai	*Klebsiella oxytoca*	+					OXY-2
4	2-3		*Pseudomonas* spp.	+		+	+		ND
5	2-5		*Klebsiella oxytoca*	+					OXY-2
6	2-6		*Pseudomonas* spp.	+		+	+		ND
7	3-1	Carrot	*Aeromonas* spp.	+					ND
8	3-2		ND	+					ND
9	3-3		*Enterobacter* spp.	+					AmpC (ACT) ACT-51v
10	4-1	Leaf lettuce	ND	+					ND
11	4-2		*Enterobacter* spp.	+					AmpC (ACT) ACT-2v
12	4-3		*Aeromonas* spp.	+					AmpC (MOX) MOX-12v
13	4-4		ND	+					ND
14	5-1	Italian parsley	ND	+					ND
15 ^1^	5-4		*Klebsiella pneumoniae*	+	+	+	+	+	SHV, TEM, CTXM, *aac(6’)-Ib*, OXA-9 (NDM-1)
16	6-1	Oka hijiki	*Pseudomonas* spp.	+			+		ND
17	6-2		ND	+					ND
18	6-3		*Pseudomonas* spp.	+			+		ND
19	7-1	# Banana 1	*Klebsiella pneumoniae*	+					SHV (SHV-27)
20	7-2		*Klebsiella variicola*	+					SHV (LEN-19v)
21	7-4		*Klebsiella variicola*	+					SHV (LEN-19v)
22^1^	8-1	Baby leaf mix	*Klebsiella pneumoniae*	+	+	+	+	+	SHV, TEM, CTX-M, *aac(6’)-Ib*,OXA-9 (NDM-1)
23^1^	8-2		*Acinetobacter baumannii*	+	+	+	+	+	AmpC (ADC) ADC-25, *intI1*
24	8-3		*Pseudomonas* spp.	+			+		ND
25	9-1	Salad mix	ND	+					ND
26	9-2		*Kosakonia* spp.	+	+	+			ND
27	9-3		*Enterobacter* spp.	+					AmpC (ACT) CMAEv
28	9-5		ND	+					ND
29	9-6		*Enterobacter* spp.	+					AmpC (ACT) CMAEv
30	10-1	Red spinach	*Enterobacter* spp.	+					AmpC (ACT) CMAEv
31	11-1	Radish	*Klebsiella oxytoca*	+					OXY-1nv
32	11-2		*Enterobacter* spp.	+			+	+	AmpC (ACT) CMAEnv
33	12-1	Mizuna	ND	+					ND
34	12-2		ND	+					ND
35	13-1	# Banana 2	*Acinetobacter* spp.	+			+		AmpC (ADC) ADCnv
36	13-2		*Enterobacter* spp.	+			+	+	AmpC (ACT) ACT-9
37	14-1	Tomato	ND	+					ND
38	14-2		ND	+					ND
39	14-3		ND	+					ND
40	15-1	Green paprika	*Klebsiella variicola*	+					SHV (LEN-27v)
41	15-2		*Proteus vulgaris*	+			+		CTX-Mw HugA
42	15-3		*Acinetobacter pittii*	+			+		AmpC (ADC) ADCnv
43	16-1	Eggplant	*Klebsiella variicola*	+					SHV (LEN-25)
44	16-2		*Acinetobacter baumannii*	+			+		AmpC (ADC) ADCnv
45	16-3		*Klebsiella pneumoniae*	+					SHV (SHV-11v, BSBL)
46	17-1	Cucumber	*Klebsiella variicola*	+					SHV (LEN-16)
47	17-2		*Klebsiella oxytoca*	+					CTX-M (OXY-2)
48	18-1	Onion	*Kosakonia* spp.	+					ND
49	18-2		ND	+					ND
50	19-1	Potato	*Klebsiella* spp.	+					OXY-1-2
51	19-2		*Klebsiella variicola*	+					SHV (LEN-13)
52	19-3		*Klebsiella aerogenes*	+					AmpC (ND) CMAEv
53	20-1	Sweet green pepper	*Klebsiella pneumoniae*	+					SHV (SHV-11)
54	21-1	Purple pepper	*Klebsiella* spp.	+					OXY-6nv
55	21-2		*Serratia marcescens*	+			+		ND

^1^ Detailed analysis of these 3 isolates by complete genome sequencing can be found in reference [19]. ^2^ ND indicates not determined. ^3^ The (+) sign indicates good growth on the corresponding medium. ^4^ The letters (v) and (nv) after β-lactamase indicate a variant and novel variant, respectively. CMAE indicates CMY2/MIR/ACT/EC family plasmid-mediated AmpC β-lactamases.

**Table 8 antibiotics-12-00090-t008:** Features of the Gram-negative isolates grown on MacConkey plates containing 100 μg/mL ampicillin from 27 non-organic (ANO: 54 isolates) vegetables and fruits.

No.	Strain ANO	Source (# Imported)	Identification by 16S rRNA Sequencing ^1^	Growing on LB Agar Medium Supplemented with ^2^					Genotype Eicosa (Octa) Plex PCR ^3^
				AMP 100 24 h	MER 4 24 h	MER 1 24 h	CTX 4 24 h	CAZ 16 24 h	
1	1-1	Apple 1	*Enterobacter* spp.	+		+			AmpC-CS (ACT) ACT-16
2	1-2		*Enterobacter* spp.	+		+			AmpC-CS (ACT) ACT-16
3	2-1	Persimmon 1	*Pseudomonas* spp.	+	+	+	+		ND
4	3-1	Tomato1	*Pseudomonas* Spp.	+		+	+		ND
5	4-1	Cucumber 1	*Klebsiella pneumoniae*	+		+			SHV-41v
6	4-2		ND	+					ND
7	5-1	Grape	*Enterobacter* spp.	+					AmpC-CS (ACTw) ACT-61v
8	5-2		*Enterobacter* spp.	+					AmpC-CS (ACTw) ACT-61v
9	6-1	Shimeji	*Klebsiella michiganensis*	+		+			AmpC-CS (ACT) CMAEv
10	6-2		*Enterobacter* spp.	+					AmpC-CS (ND) CMAEnv
11	7-1	Hakusai	*Enterobacter* spp.	+	+	+	+		AmpC-CS (ACT) ACT-64, *intI1*
12	7-2		*Pseudomonas alcaligenes*	+	+	+	+	+	ND
13	8-2	Apple 2	*Citrobacter freundii*	+			+	+	AmpC-CS (ND) CMAEnv
14	9-1	Pear 1	*Pseudomonas putida*	+		+			ND
15	9-2		ND	+		+			ND
16	9-4		ND	+		+			ND
17	9-5		ND	+		+			ND
18	10-2	Persimmon 2	*Enterobacter* spp.	+			+	+	AmpC-CS (ACT) CMAE
19	10-3		*Citrobacter* spp.	+			+	+	AmpC-CS (ND) CMAEnv
20	11-1	# Kiwi	*Pseudomonas* spp.	+			+	+	AmpC-CS (ND) AmpCnv
21	11-3		*Rahnella aquatilis*	+			+		CTX-M (RANH-2, ESBL)
22	12-1	Pear 2	*Kosakonia* spp.	+					ND
23	12-2		*Kosakonia* spp.	+					ND
24	12-3		ND	+					ND
25	14-1	Tomato 2	*Pseudomonas* spp.	+		+	+		ND
26	14-2		*Pantoea ananatis*	+			+		ND
27	14-3		*Pantoea ananatis*	+		+			ND
28	17-1	Potato 1	*Klebsiella pneumoniae*	+					SHV-1v
29	18-1	Cucumber 2	*Enterobacter cloacae*	+			+	+	AmpC-CS (ACT) CMAE
30	18-2		*Enterobacter* spp.	+			+		TEM-1, AmpC-CS (ACT) ACT-32, *intI1*
31	18-3		*Klebsiella pneumoniae*	+					SHV-1v
32		Single	*Enterobacter* spp.	+					AmpC-CS (ACT) AZECL-32
33	19-1	Tomato 3	ND	+					ND
34	19-2		ND	+					ND
35	19-3		*Enterobacter* spp.	+					AmpC-CS (ACT) ACT-32
36	20-2	Potato 2	*Pseudomonas* spp.	+			+		ND
37	20-3		*Pseudomonas* spp.	+			+		ND
38	21-1	Green paprika 1	ND	+					ND
39	21-2		ND	+					ND
40	21-3		*Pseudomonas fulva*	+		+	+		ND
41	21-4		*Pseudomonas fulva*	+		+	+		ND
42	22-1	Korean lettuce	*Enterobacter* spp.	+		+	+	+	AmpC-CS (ACT) ACT-51v
43	22-3		*Pseudomonas* spp.	+		+	+		ND
44	23-1	Lettuce	*Klebsiella pneumoniae*	+		+			SHV-1
45	24-1	# Gold kiwi	*Enterobacter cloacae*	+			+	+	AmpC-CS (ACT) CMAE
46	24-2		*Enterobacter cloacae*	+			+	+	AmpC-CS (ACT) CMAE
47	24-3		*Enterobacter cloacae*	+					AmpC-CS (ACT) CMAE
48	25-1	Persimmon 3	ND	+					ND
49	25-2		*Klebsiella pneumoniae*	+					SHV-1v
50	25-3		ND	+					ND
51	26-2	Green paprika 2	*Pseudomonas* spp.	+	+	+	+		ND
52	26-3		*Pseudomonas* spp.	+	+	+	+		ND
53	27-1	Paprika	*Enterobacter cloacae*	+			+	+	AmpC-CS (ACT) CMAE
54	27-2		*Pseudomonas putida*	+	+	+	+		ND

^1^ ND indicates not determined. ^2^ The (+) sign indicates good growth on the corresponding medium. ^3^ The letters (v) and (nv) after β-lactamase indicate a variant and novel variant, respectively. CMAE indicates CMY2/MIR/ACT/EC family plasmid-mediated AmpC β-lactamases.

## Data Availability

Data is contained within the article.

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
