# Peer review of "A Novel Single-Tube Eicosaplex/Octaplex PCR System for the Detection of Extended-Spectrum β-Lactamases, Plasmid-Mediated AmpC β-Lactamases, and Integrons in Gram-Negative Bacteria"

_antibiotics, 2023, doi:10.3390/antibiotics12010090_

Round 1

Reviewer 1 Report

In this article, the authors demonstrate the utility of two self-developed multiplex PCR tests for the detection of ESBLs, plasmid-mediated AmpC β-lactamases, and integrons in gram-negative bacteria isolates from non-organic (ANO) and organic (AO) vegetables 24 and fruits. The methodological approach seems optimal as well as the design of specific primers forfor eicosaplex and octaplex PCRs. The gram negative bacteria used in this study as positive controls should, in my opinion, be representative of a greater number of regions since all microorganisms clinically relevant to humans were isolated in Egypt as well as those of plant, animal and or environmental are from Japan (see table 1). The conclusions should be discussed more broadly, not only reiterating the methodological aspect of the study

Author Response

The authors would like to thank Reviewer #1 very much for reviewing and commenting on our manuscript. Due to the limited number of region that we are isolating bacteria from it (i.e., exclusively from Egypt and Japan), we used this composition of positive controls. However, in the next coming manuscripts, we will try to contact several researchers around the world and bring gram-negative bacteria for the evaluation purpose.

Conclusions were improved by adding the following statements:

“This multiplex PCR assay is a promising workflow in bacterial diagnosis in high-risk patients providing a great assistance in optimizing and fast choice of antibiotics for treating infections due to ESBLs- and AmpC β-lactamases-producing gram-negative pathogens.”

Reviewer 2 Report

Dear Editor and authors

The article on title “A Novel Single-Tube Eicosaplex/Octaplex PCR System for the Detection of Extended-Spectrum β-lactamases, Plasmid-Mediated AmpC β-lactamases, and Integrons in Gram-Negative Bacteria”, Is really interesting and this manuscript is well performed and easy to follow. I think that this adds important results to be consider for publishing in this journal. So I recommend for acceptance in present form.

Author Response

The authors would like to thank Reviewer #2 very much for reviewing and commenting on our manuscript.

Reviewer 3 Report

Soliman et al. developed two multiplex PCR assays for screening clinically relevant β-lactamases and integrons in gram-negative bacteria. They applied the method to gram-negative isolates from ANO and AO vegetables and fruits. They were able to identify various β-lactamase and integrase genes/variants in these isolates. I enjoyed the neat gel figure in Fig. 3C. The manuscript is overall in its good shape. My only comment is in line 41, where the description of "antibiotics that intend to kill them" is inaccurate. Not all antibiotics kill bacterial cells (i.e., cidal), quite a lot inhibits bacterial growth (i.e., static).

Author Response

The authors would like to thank Reviewer #1 very much for reviewing and commenting on our manuscript. We have modified the statement as follows: “……to overcome the action of the antibiotics that intend to kill them (i.e., bactericidal), or inhibit their growth (i.e., bacteriostatic)”.